# The Effect of Hospital-Based Liquid Diet and Commercial Formulas on Laboratory Parameters and Postoperative Complications in Patients with Head and Neck Cancer

**DOI:** 10.3390/jcm13071844

**Published:** 2024-03-22

**Authors:** Aldona Chloupek, Dariusz Jurkiewicz

**Affiliations:** 1Department of Cranio-Maxillofacial Surgery, Military Institute of Medicine—National Research Institute, 04-141 Warsaw, Poland; 2Department of Otolaryngology and Oncology, Military Institute of Medicine—National Research Institute, 04-141 Warsaw, Poland; djurkiewicz@wim.mil.pl

**Keywords:** commercial formulas, enteral nutrition, head and neck cancer, hospital-based diet, malnutrition

## Abstract

**Background:** Patients with head and neck cancer (HNC) are at high risk of malnutrition. The aim of this study was to compare the effect of polymeric formulas available commercially and a high-protein liquid diet prepared in the hospital on laboratory parameters and postoperative complications in patients undergoing surgery for HNC. **Methods:** This single-center retrospective study included 149 patients who underwent surgery for HNC between 2008 and 2017. The following data were collected: patient and tumor characteristics, postoperative complications, and laboratory parameters measured at baseline and after surgery, including creatinine, alanine transaminase (ALT), aspartate transaminase (AST), and blood glucose levels. Correlations between the duration of enteral nutrition and blood parameters were assessed. **Results:** After surgery, patients receiving commercial formulas had lower creatinine and blood glucose levels and higher ALT and ASP levels than those on the hospital-based diet. The longer duration of feeding with commercial formulas before surgery was associated with enhanced preoperative levels of ALT and ASP and with lower postoperative blood glucose. Patients on the hospital-based diet had a higher rate of postoperative complications than those receiving commercial formulas (16.1% vs. 3.3%). **Conclusions:** There were no clinically important differences in blood parameters among patients with HNC depending on the type of preparations used for enteral feeding. However, increased levels of liver enzymes in patients fed with commercial formulas were notable. The early initiation of enteral nutrition before surgery helped achieve normal blood glucose levels after surgery. The use of commercial preparations contributed to reducing the number and incidence of postoperative complications.

## 1. Introduction

Malnutrition, a significant loss of weight and body resources, leading to poorer quality of life and prognosis, is commonly observed in patients with cancer [1]. Cancer-associated malnutrition results from a combination of reduced food intake and host-derived or tumor-induced metabolic dysregulation (e.g., elevated resting metabolic rate, insulin resistance, lipolysis, proteolysis) caused by systemic inflammation and catabolic mediators [2,3]. In advanced disease, it may progress to cancer cachexia—a multifactorial syndrome of progressive and unintentional body weight loss and muscle wasting, which occurs in 30% of all cancer patients and is a strong predictor of poor survival [1,3,4]. Malnutrition is associated with enhanced treatment toxicity, poorer compliance with treatment, lower quality of life, worse clinical outcomes, an increased rate of postoperative complications, and longer hospitalization [3]. Moreover, it may account for 20% of cancer deaths [3]. The prevalence of malnutrition in patients with cancer varies depending on the tumor type and stage, the treatment type and setting, and comorbidities, reaching up to 80% of patients [3,5]. Risk groups among the cancer population include patients with head and neck cancer (HNC), gastrointestinal cancer (e.g., pancreatic, esophageal, and gastric cancers), and lung cancer [5].

Head and neck cancers are the seventh most common type of malignancy worldwide and represent 6% of all cancers, with nearly 1 million new diagnoses each year [6,7]. In 90% of cases, HNCs develop from the squamous epithelium in the oral cavity, pharynx, and larynx. In the HNC population, at the time of diagnosis, an increased risk of malnutrition was reported in 28.6% to 67% of patients and overt malnutrition in 23.8% to 48.9% of patients [5,8]. Malnutrition in HNC is usually defined as unintended weight loss of more than 5% to 10% during the previous 1 to 6 months and a body mass index (BMI) of less than 18.5 to 20 kg/m^2^. It develops not only because of higher tumor-related nutritional requirements, but also because of the tumor location and the side effects of treatment [7,9]. The physical presence of a tumor may make food consumption difficult due to impaired swallowing, thus contributing to reduced food intake even before the treatment is started [10]. Surgical tumor resection used for early-stage HNC may lead to anatomical changes and postoperative complications that affect the oral intake of nutrients and often require enteral nutrition [9]. In addition, the patient has increased nutritional requirements to support recovery and wound healing [9]. Moreover, radiotherapy and chemotherapy, which are commonly used in HNC treatment, are associated with complications such as dysphagia, odynophagia, oral mucositis, xerostomia, dysgeusia, pain, fatigue, gastrointestinal problems, and loss of appetite, all of which can increase the risk of malnutrition [9,10]. Finally, some social factors can also worsen the patient’s nutritional status, including poor dietary habits at baseline, restrictions on dietary choices, lack of social support, depression, and high alcohol consumption [9]. Silva et al. [11] reported that risk factors for malnutrition in patients with HNC included the tumor’s location in the oral cavity and oropharynx, younger age, lower education status, and a BMI lower than 18.5 at diagnosis.

It was shown that nutritional support for HNC patients at any stage of treatment prevents cancer-associated malnutrition by improving the quality of life, physical performance, metabolism, and tolerance of cancer treatment [1,12]. A balanced diet, with an emphasis on a high-caloric and high-protein diet, is essential for HNC patients to meet their nutritional requirements [2,12]. Oral nutritional supplementation involves the use of commercially available liquid or semi-solid enteral formulas that provide the necessary macronutrients and micronutrients. It can be helpful in patients with mild and moderate malnutrition and dysphagia [12]. Enteral nutrition is often required in HNC patients with swallowing problems. It is recommended when oral feeding constitutes less than 60% of the individual nutritional requirement for more than 2 weeks and is inadequate to maintain body weight, or when the interruption of oral feeding lasts more than 7 days [2,7,12,13]. For short-term enteral feeding of up to 4 weeks, nasogastric or nasojejunal tubes can be used, while long-term enteral nutrition requires the use of percutaneous endoscopic gastrostomy (PEG) and jejunostomy tubes [12,13]. Enteral feeding formulas can be classified into polymeric, monomeric, and disease-specific ones [8,14]. Polymeric formulas, which contain carbohydrate polymers, complete proteins, and triglycerides, are suitable for most patients, while monomeric formulas are considered more appropriate for patients with digestive dysfunction [8,14]. Although commercially available enteral formulas provide complete nutritional supplementation in terms of vitamin, protein, and caloric requirements, less expensive home-based recipes using nutrient-dense ingredients may be equally sufficient [8]. Early enteral nutrition (prior to surgery) was shown to improve wound healing and reduce the length of hospitalization in patients with malnutrition [8,12]. Parenteral nutrition is only considered if enteral nutrition is not possible or contraindicated [12,13].

The aim of this study was to compare polymeric enteral formulas that are commercially available with a high-protein liquid diet prepared in the hospital in terms of their effect on (1) blood parameters measured before and after surgery; and (2) the occurrence of treatment-related complications in patients with HNC.

## 2. Materials and Methods

### 2.1. Study Design and Participants

In this single-center retrospective study, we collected data from 149 patients who underwent surgery for HNC (primarily cancer of the tongue and the floor of the mouth) between 2008 and 2017 at the Department of Cranio-Maxillofacial Surgery in the Military Institute of Medicine in Warsaw, Poland. The inclusion criteria were as follows: C01 diagnosis (extensive surgery of oral cavity, pharyngeal, and laryngeal cancer with reconstruction) according to the Polish patient classification scheme (JGP)—a national diagnosis-related group (DRG) system [15]; surgical method (tumor resection, resection of neck lymph nodes, or reconstruction); lack of perioperative blood transfusion and albumin administration; and enteral feeding in the perioperative period. Participants were divided into 2 groups: a group of 87 patients who received a high-protein liquid diet prepared in the hospital and a group of 62 patients who were fed with commercial polymeric formulas.

The following data were obtained from medical records: the patient’s age, sex, weight, and height; length of hospital stay; stage, location, size, and histopathological features of the resected tumor; type of resection and reconstructive surgery; occurrence of postoperative complications, and the type of feeding (based on subjective global assessment [16,17] and patient diet cards). In addition, the following laboratory parameters measured at baseline and after surgery and were compared: red blood cell (RBC) count, white blood cell (WBC) count, hemoglobin, hematocrit, mean corpuscular volume (MCV), creatinine, albumin, total protein, low-density lipoprotein cholesterol (LDL-C), high-density lipoprotein cholesterol (HDL-C), triglycerides, alanine transaminase (ALT), aspartate transaminase (AST), and blood glucose.

The study was approved by the Bioethics Committee at the Military Institute of Medicine (17/WIM/2017; decision from: 15 March 2017). Informed patient consent was not required due to the retrospective design of the study.

### 2.2. Perioperative Nutrition

In line with the principles of the proper nutrition of hospitalized patients in Poland [18], the high-protein liquid diet contained 115 g of protein (80 g of animal protein and 35 g of plant protein) with an energy value of 2000 kcal per day. The hospital-based blenderized high-protein liquid diet was administered as a bolus of 350 to 500 mL 3 times a day, using a nasogastric tube.

The remaining patients received commercial polymeric formulas. On the first day of enteral feeding, they were administered a standard polymeric diet. From the second day onward, they received a high-calorie and/or high-protein diet, depending on their albumin levels, body weight, and BMI. In patients with a BMI equal to or lower than 23 kg/m^2^, the energy value of the diet was calculated as 35 to 45 kcal/kg of current body weight per day. In obese patients with a BMI higher than 23 kg/m^2^, the energy value was calculated considering the ideal body weight. The protein requirement was estimated at 2 to 3 g/kg of body weight per day. Protein content and energy value were calculated according to Polish Recommendations on Enteral and Parenteral Nutrition in Oncology [19] and European Society for Parenteral and Enteral Nutrition guidelines for cancer patients in the perioperative period [20]. Commercial formulas were administered as a bolus of 200 to 300 mL 5 to 6 times a day, using a nasogastric tube or PEG under the strict monitoring of gastric retention.

### 2.3. Statistical Analysis

Quantitative variables were presented as an arithmetic mean with standard deviation (SD) or a median with minimum and maximum values (range). Qualitative variables were presented as the number of observations and percentage. Significant differences between the means in 2 groups were determined using Student’s *t*-test and paired Student’s *t*-test. The significance of differences between more than 2 groups was assessed using a 1-way analysis of variance. To establish the direction and strength of a relationship between variables, correlation analysis was performed, and the Pearson correlation coefficient (r) was estimated. The Kolmogorov–Smirnov test and the Lilliefors test were used to test the normality of the distribution of variables. A *p* value of less than 0.05 was considered statistically significant. All statistical analyses were performed using STATISTICA 6.0 PL (data analysis software system) (Statsoft, Tulsa, OK, USA).

## 3. Results

### 3.1. Characteristics of Patients

The study included 149 patients: 90 men (60.4%) and 59 women (39.6%). The mean age of the patients was 64.5 years (range: 29–91 years). The basic characteristics of the study groups are presented in Table 1. The median BMI was within a reference range in both groups, but it was higher in patients on the hospital-based diet vs. those receiving commercial formulas (23 vs. 21 kg/m^2^). The lowest BMI (corresponding to malnutrition) was 17.2 kg/m^2^ in the group on the hospital-based diet and 15.2 kg/m^2^ in the group receiving commercial formulas. The highest BMI (corresponding to obesity) was about 39 kg/m^2^ in both groups. The length of hospital stay was similar in both groups (Table 1). The longest duration of hospital stay was 58 days.

On histopathology, the most common type of cancer in both groups was keratotic squamous cell carcinoma (Table 1). In more than 80% of patients, the tumor was classified as intermediate grade (G2). The most common tumor sites were the tongue and/or floor of the mouth, followed by the gums (Table 1). The median tumor size was 7 cm^2^ in patients on the hospital-based diet and almost 9 cm^2^ in those receiving commercial products (Table 1).

In the group fed with commercial formulas, a nasogastric tube was used in 64.5% of patients and PEG in 33.9%. In the remaining 1.6% of patients, both methods were used to provide nutrition, first a nasogastric tube and then PEG (Table 1). Almost all patients (n = 59) in this group received enteral feeding already before surgery. The average duration of enteral nutrition before surgery was 2.5 days ± 2.1 (median, 2; range, 0–9). During the hospital stay, enteral nutrition was maintained for 13.7 days ± 8.7 (median, 11; range, 6–58).

### 3.2. Impact of Enteral Nutrition on Laboratory Parameters

After surgery, there was a significant reduction in RBC count and blood hemoglobin, hematocrit, albumin, total protein, potassium, and sodium levels in both groups. On the other hand, there was an increase in WBC count, triglycerides, AST, and blood glucose levels (Table 2). Additionally, in patients receiving commercial formulas, there was a reduction in creatinine and HDL levels and an increase in ALT levels.

A comparison of laboratory parameters between groups showed that patients receiving commercial formulas had higher WBC counts and lower MCV and total protein levels before surgery than patients on the hospital-based diet (Table 2). After surgery, the mean values of creatinine levels were lower (0.7 ± 0.2 mg/dL vs. 1.1 ± 1.4 mg/dL, *p* = 0.030) and those of liver enzymes were higher (ALT, 25.6 ± 19.9 U/L vs. 15.4 ± 7.5 U/L, *p* < 0.001; ASP, 30.7 ± 16.4 U/L vs. 25.6 ± 14.4 U/L, *p* = 0.049) in patients receiving commercial formulas vs. those receiving the hospital-based diet. The mean blood glucose levels were significantly higher in patients on the hospital-based diet than in those fed with commercial polymeric formulas (147.1 ± 49.0 mg/dL vs. 125.2 ± 58.1 mg/dL, *p* = 0.014). Percentage changes in laboratory parameters after surgery in patients on a high-protein liquid diet prepared in the hospital versus those on commercial polymeric formulas are presented in Figure 1.

### 3.3. Correlations between Tumor Size and Blood Parameters

The analysis of correlations between tumor size and laboratory parameters showed that larger tumors were correlated with lower RBC count (r = −0.27, *p* = 0.013), hemoglobin (r = −0.36, *p* = 0.001), and hematocrit (r = −0.23, *p* = 0.032) levels before surgery in patients on the hospital-based diet. In patients fed with commercial products, there was a moderate positive correlation between tumor size and ALT (r = 0.31, *p* = 0.013) and AST (r = 0.038, *p* = 0.003) levels. Detailed data are presented in Table 3.

There was a weak inverse correlation between tumor size and albumin (r = −0.026, *p* = 0.018) and ALT (r = −0.23, *p* = 0.036) levels after surgery in patients on the hospital-based diet (Table 3). In the other group, there was a positive correlation between tumor size and LDL cholesterol levels (r = 0.33; *p* = 0.008) after surgery.

### 3.4. Correlations between Duration of Enteral Nutrition and Laboratory Parameters

We assessed correlations between the duration of enteral nutrition (nutrition before surgery and, separately, the total duration of nutrition) and laboratory parameters (before and after surgery) in patients fed with commercial formulas. The longer duration of nutrition was correlated with ALT levels before surgery (r = 0.26, *p* = 0.038) (Table 4, Figure 2). The correlation was stronger when only the duration of nutrition before surgery was considered (r = 0.39, *p* = 0.002). A similar correlation was noted for the duration of nutrition before surgery and AST levels (r = 0.29, *p* = 0.025).

Our analysis showed that the longer total duration of enteral nutrition (both before surgery and in the perioperative period) as well as the longer duration of nutrition before surgery were associated with a significant reduction in WBC count and hemoglobin and albumin levels after surgery in patients fed with commercial formulas (Table 4, Figure 2). Moreover, the RBC count was inversely correlated with the duration of feeding before surgery (r = −0.26; *p* = 0.044). There was also a significant correlation between the longer duration of nutrition before surgery and reduced blood glucose levels after surgery (Table 4, Figure 2).

### 3.5. Postoperative Complications Depending on the Type of Enteral Feeding

Most postoperative complications occurred in patients on the hospital-based diet. The rate of complications in these patients was 16.1% vs. 3.3% in patients fed with commercial products. The most common complications were postoperative wound necrosis and oral cutaneous fistulas. One case of gastrointestinal perforation following PEG tube insertion was reported in patients receiving commercial formulas. In the group on the hospital-based diet, four patients (4.6%) died after surgery due to poor general condition. Among patients fed with commercial formulas, one death was reported in a patient with postoperative bleeding from the oral cavity and fever. Detailed data on complications in both groups are presented in Table 5.

## 4. Discussion

It is widely accepted that the appropriate nutritional support of cancer patients increases treatment tolerance, reduces complication rates and the length of hospital stay, improves clinical outcomes and the quality of life, and lowers healthcare costs [3]. However, the use of blenderized tube feeding vs. commercial formulas for enteral nutrition is still debatable [21,22]. By the 1970s, blenderized tube feeding was gradually replaced by commercial formulas—sterile products with a known nutrient composition—due to their ease of administration; reduced labor expenses; and less concern about sanitation, osmolarity, and viscosity [21,23]. Although the advantages of commercial formulas are commonly accepted, healthcare providers and patients increasingly recognize the benefits of eating blended fresh foods and liquids. This type of feeding seems to be more physiological compared with highly processed and monotonous feeding substrate [21,24]. Carter et al. [23] claimed that concerns about the variability, microbial load, and costs of blenderized tube feeding may be overstated [23]. Blended diets were reported to have numerous advantages including improved gastrointestinal tolerance, the prevention of weight loss, support for the growth of nonpathogenic foodborne bacteria that are beneficial for health, and reduced healthcare costs [21,24]. The advantages of blenderized tube feeding are also well documented in pediatric populations; however, studies in adults are scant [24].

In this study, we compared commercial polymeric formulas and a blenderized high-protein liquid diet prepared in the hospital in terms of their effect on hematological and biochemical parameters in HNC patients after surgery. Laboratory blood tests, including hemoglobin as a marker of anemia, lymphocyte count, albumin, and glucose, are widely used in the assessment of patients with malnutrition [25]. Moreover, hemoglobin levels, leukocyte, neutrophil, monocyte, and platelet counts, and neutrophil-to-lymphocyte, platelet-to-lymphocyte, and lymphocyte-to-monocyte ratios were found to be associated with the overall survival of HNC patients, depending on the tumor site [26,27,28,29]. To our knowledge, there has been only one study that assessed the impact of enteral nutrition on blood parameters within a year from starting treatment at a nutrition clinic. However, there was no control group, and only 45% of the population had cancer [25]. Another study assessed changes in hemoglobin, albumin, and phosphate levels, among other parameters, in children receiving commercial nutritional formulas or blenderized tube feeds. No significant differences in biochemical parameters between groups were reported [30]. This is in line with our study, in which the observed reductions in RBC, hemoglobin, hematocrit, albumin, total protein, potassium, and sodium levels were not significantly different between groups receiving different types of nutrition. However, there was a notable increase in the levels of liver enzymes (ALT, AST) after surgery in patients receiving commercial formulas. 

Another interesting finding is a positive correlation between liver enzymes and tumor size in patients who received commercial formulas. It is known that patients with HNC are particularly susceptible to liver disease because of the high prevalence of alcohol abuse, which is one of the major risk factors for cancer of the upper aerodigestive tract [31]. Therefore, further research is required to more precisely assess the relationship between tumor size and enhanced level of liver enzymes in HNC.

Patients fed with the blenderized liquid diet showed high glucose levels after surgery. Hyperglycemia is known to be responsible for excessive glucose supply to cancer cells, resulting in tumor growth; cancer progression; and increased resistance to, or intolerance of, chemotherapy [32]. However, it might be hypothesized that lowering the carbohydrate content of the hospital-based diet could help reduce this effect [33].

Our study showed that the longer duration of enteral nutrition in patients fed with commercial formulas was associated with higher ALT and AST levels after surgery, especially when enteral nutrition was initiated before surgery. This may confirm the negative effect of commercial formulas on the liver described above. In contrast, Konecka et al. [25] described a significant decrease in liver enzyme activity within 1 year from the initiation of nutritional treatment; however, the authors did not provide data on the type of enteral nutrition or any surgical treatment applied at that time. Similar discrepancies were also noted for other parameters. In our study, the duration of enteral nutrition in patients fed with commercial formulas was inversely correlated with hemoglobin and albumin levels. On the other hand, Konecka et al. [25] reported improved complete blood count parameters and stable albumin levels during 1 year of tube feeding. This discrepancy may be due to a different duration of enteral nutrition between our studies (2 weeks vs. 1 year). As for the inverse correlation between the WBC count and the duration of enteral nutrition in our study, it can be assumed that an elevated WBC count after surgery decreases and inflammation is gradually reduced with the longer duration of enteral feeding. This is in line with the results reported by Konecka et al. [25].

In our study, the early introduction of enteral nutrition had a beneficial effect on blood glucose levels after surgery. However, surprisingly, the RBC count decreased with a longer duration of preoperative enteral feeding with commercial formulas. According to Bossola [34], the prophylactic feeding approach does not offer significant advantages in terms of nutritional outcomes, interruptions of radiotherapy, and survival in HNC patients, as compared with the reactive feeding approach. Brown et al. [35] reported no effect of early tube feeding via prophylactic gastrostomy on weight loss, quality of life, and clinical outcomes. According to Akbulut [36], at least 10 days of nutritional support are recommended before major digestive surgery in patients with severe malnutrition, even when surgery has to be delayed. However, an early approach to enteral nutrition should not be used routinely in all patients [36].

The correlation between the degree of malnutrition and elevated risk of postoperative complications is well established [36]. In our study, the number and frequency of postoperative complications in patients on the hospital-based diet were higher than those in patients fed with commercial formulas, which may indicate that commercial formulas can be beneficial in improving postoperative wound healing. However, there were no significant differences between groups in terms of the length of hospital stay.

### Limitations

Our study has several limitations. First, it was a retrospective study, which makes it difficult to draw definitive conclusions. Second, data were collected from 2008 to 2017. During that period, new surgical techniques were introduced that shortened the duration of surgical procedures, which may have affected the length of hospital stay. Third, over subsequent years, patients with a more advanced stage of cancer were referred for surgery, which may have affected the homogeneity of the study population. Finally, we did not perform a sample size analysis. A too-small sample size may be associated with a large number of results that are not statistically significant. Conversely, the nonsignificant differences regarding laboratory parameters, which did not vary significantly between the study groups, would require a sample size of several thousand to demonstrate significance in such comparisons, a requirement that is impossible to meet under our conditions. Nevertheless, this analysis is based on all patients who underwent surgery for HNC in our department and met the inclusion criteria over a period of nine years.

## 5. Conclusions

Patients with HNC are particularly susceptible to nutritional problems due to the burden of the disease and consequent inflammation. Nutritional support could improve clinical outcomes and minimize the risk of postoperative complications, thus improving survival and the quality of life. However, it is unclear if commercial polymeric formulas are superior to blenderized tube feeding used for enteral nutrition. Generally, our study showed no clinically important differences in hematological and biochemical parameters in HNC patients after surgery depending on the type of feeding. However, the increase in liver enzymes in patients fed with commercial preparations is alarming and requires further research. The early approach to enteral nutrition had a positive effect on blood glucose levels after surgery. Finally, while the use of commercial formulas did not shorten the length of hospital stay, it seems to contribute to reducing the number and incidence of postoperative complications in HNC patients compared to the hospital-based diet.

## Figures and Tables

**Figure 1 jcm-13-01844-f001:**
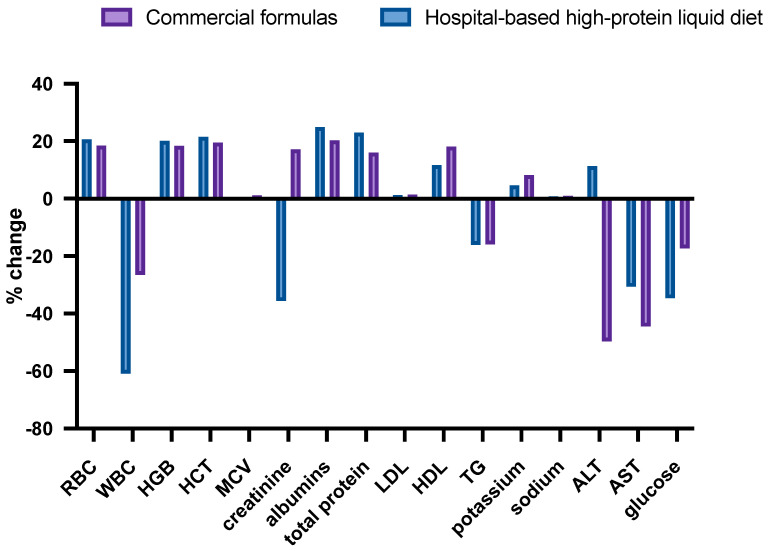
Changes in laboratory parameters after surgery in patients on a high-protein liquid diet prepared in the hospital versus those on commercial polymeric formulas.

**Figure 2 jcm-13-01844-f002:**
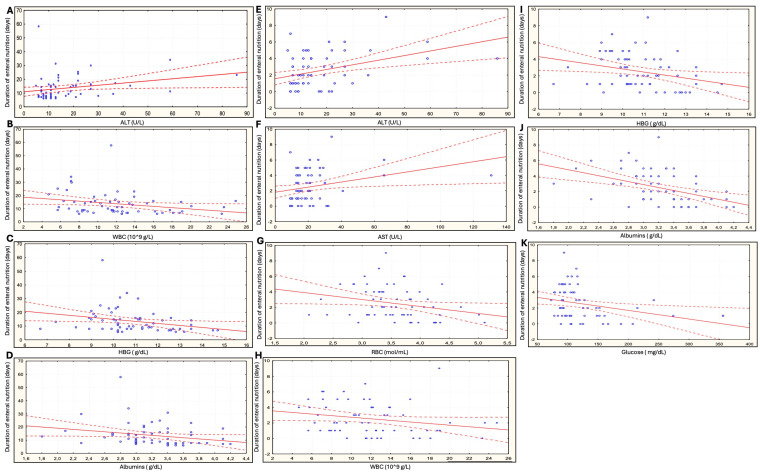
Correlations between the duration of enteral nutrition and laboratory parameters in patients fed with commercial polymeric formulas. (**A**–**D**) Before surgery and in the perioperative period. (**E**–**K**) Before surgery. ALT—alanine transaminase, AST—aspartate transaminase, HGB—hemoglobin, NS—not significant, RBC—red blood cell, WBC—white blood cell.

**Table 1 jcm-13-01844-t001:** Basic characteristics of patients receiving high-protein liquid diet prepared in the hospital and those receiving commercial polymeric formulas.

Parameter	Hospital-Based High-Protein Liquid Diet (*n* = 87)	Commercial Formulas (*n* = 62)
Sex		
Women	35 (40.2)	24 (38.7)
Men	52 (59.8)	38 (61.3)
Age (years)	67 (29–91)	61 (34–85)
BMI (kg/m^2^)	23.0 (17.2–39.3)	21.2 (15.2–39.1)
Length of hospital stay (days)	10 (3–58) 12.9 ± 9.4	11 (6–58) 13.6 ± 8.7
Histopathology		
Keratotic squamous cell carcinoma	82 (94.2)	55 (88.7)
Squamous cell carcinoma without keratosis	3 (3.4)	4 (6.5)
Other	2 (2.2)	3 (4.8)
Tumor grade		
G1	6 (6.9)	5 (8.)
G1/G2	4 (4.6)	2 (3.2)
G2	73 (83.9)	51 (82.3)
G2/G3	1 (1.1)	–
G3	3 (3.4)	4 (6.5)
Tumor location		
Tongue and floor of the mouth	18 (20.6)	18 (29.0)
Tongue	16 (18.4)	10 (16.1)
Floor of the mouth	16 (18.4)	15 (24.2)
Gums	9 (10.2)	9 (14.5)
Other	28 (32.4)	10 (16.2)
Tumor size (cm^2^)	7.0 (0.2–55.0)	8.8 (0.5–36.0)
Type of feeding		
Nasogastric tube	87 (100)	40 (64.5)
PEG	–	21 (33.9)
Nasogastric tube/PEG	–	1 (1.6)

Data are presented as medians (min–max), mean ± SD, or n (%) of patients. BMI—body mass index, PEG—percutaneous endoscopic gastrostomy.

**Table 2 jcm-13-01844-t002:** Laboratory parameters before and after surgery in patients receiving the high-protein liquid diet prepared in the hospital and those receiving commercial polymeric formulas.

Parameter	Hospital-Based High-Protein Liquid Diet (*n* = 87)			Commercial Formulas (*n* = 62)			*p*-Value (2 Groups Compared)	
	Before Surgery	After Surgery	*p*-Value	Before Surgery	After Surgery	*p*-Value	Before Surgery	After Surgery
RBC [L]	4.3 (0.5)	3.4 (1.1)	<0.001	4.4 (0.7)	3.6 (0.6)	<0.001	NS	NS
WBC [L]	7.8 (2.5)	12.6 (4.9)	<0.001	9.6 (3.8)	12.1 (4.6)	0.001	<0.001	NS
Hemoglobin [g/dL]	13.2 (2.0)	10.5 (1.9)	<0.001	13.3 (1.9)	10.8 (1.6)	<0.001	NS	NS
Hematocrit [%]	40.1 (4.6)	31.5 (6.6)	<0.001	39.9 (5.6)	32.1 (4.8)	<0.001	NS	NS
MCV [fL]	93.8 (5.9)	93.4 (10.9)	NS	91.7 (6.2)	90.6 (5.1)	NS	0.032	NS
Creatinine [mg/dL]	0.8 (0.2)	1.1 (1.4)	NS	0.9 (0.3)	0.7 (0.2)	0.001	NS	0.030
Albumin [g/dL]	4.2 (0.6)	3.1 (0.7)	<0.001	4.0 (0.6)	3.2 (0.5)	<0.001	NS	NS
Total protein [g/dL]	7.2 (0.7)	5.5 (4.1)	<0.001	6.8 (0.9)	5.7 (0.6)	<0.001	0.004	NS
LDL-C [mg/dL]	1.4 (0.2)	1.4 (0.2)	NS	1.4 (0.2)	1.4 (0.2)	NS	NS	NS
HDL-C [g/dL]	4.1 (0.6)	3.6 (2.2)	NS	4.1 (0.6)	3.4 (0.5)	<0.001	NS	NS
Triglycerides [g/dL]	6.2 (0.5)	7.2 (0.6)	<0.001	6.2 (0.5)	7.2 (0.5)	<0.001	NS	NS
Potassium [mmol/L]	4.5 (0.6)	4.3 (0.6)	0.015	4.6 (0.5)	4.2 (0.5)	<0.001	NS	NS
Sodium [mmol/L]	140.0 (2.2)	138.8 (2.9)	0.003	139.7 (2.5)	138.3 (4.3)	0.038	NS	NS
ALT [U/L]	17.4 (8.0)	15.4 (7.2)	NS	17.1 (14.6)	25.6 (19.9)	0.008	NS	<0.001
AST [U/L]	19.6 (7.5)	25.6 (14.4)	0.001	21.2 (18.0)	30.7 (16.4)	0.003	NS	0.049
Glucose [mg/dL]	109.2 (28.2)	147.1 (8.9)	<0.001	106.7 (27.5)	125.2 (58.1)	0.026	NS	0.014

Data are presented as means (standard deviation). Significant differences at *p* < 0.05. ALT—alanine transaminase, AST—aspartate transaminase, HDL-C—high-density lipoprotein cholesterol, LDL-C—low-density lipoprotein cholesterol, MCV—mean corpuscular volume, NS—not significant, RBC—red blood cell, TG—triglycerides, WBC—white blood cell.

**Table 3 jcm-13-01844-t003:** Correlations between tumor size and laboratory parameters before and after surgery in patients receiving the high-protein liquid diet prepared in the hospital and those receiving commercial polymeric formulas.

Blood Parameters	Tumor Size			
	Hospital-Based High-Protein Liquid Diet		Commercial Formulas	
	r	*p*-Value	r	*p*-Value
RBC ^1^	−0.27	0.013	–	NS
Hemoglobin ^1^	−0.36	0.001	–	NS
Hematocrit ^1^	−0.23	0.032	–	NS
ALT ^1^	–	NS	0.31	0.013
AST ^1^	–	NS	0.38	0.003
Albumin ^2^	−0.26	0.018	–	NS
ALT ^2^	−0.23	0.036	–	NS
LDL-C ^2^	–	NS	0.33	0.008

^1^ before surgery, ^2^ after surgery. Data are presented as the Pearson correlation coefficient (r). Significant differences at *p* < 0.05. ALT—alanine transaminase, AST—aspartate transaminase, LDL-C—low-density lipoprotein cholesterol, NS—not significant, RBC—red blood cell.

**Table 4 jcm-13-01844-t004:** Correlations between the duration of enteral nutrition and laboratory parameters in patients fed with commercial polymeric formulas.

Parameter	Duration of Enteral Nutrition			
	Before Surgery and in the Perioperative Period		Before Surgery	
	r	*p*-Value	r	*p*-Value
ALT ^1^	0.26	0.038	0.39	0.002
AST ^1^	–	NS	0.29	0.025
RBC ^2^	–	NS	−0.26	0.044
WBC ^2^	−0.26	0.040	−0.23	0.046
Hemoglobin ^2^	−0.27	0.033	−0.28	0.026
Albumin ^2^	−0.24	0.050	−0.43	<0.001
Glucose ^2^	–	NS	−0.31	0.016

^1^ before surgery, ^2^ after surgery. Data are presented as the Pearson correlation coefficient (r). Significant differences at *p* < 0.05. ALT—alanine transaminase, AST—aspartate transaminase, NS—not significant, RBC—red blood cell, WBC—white blood cell.

**Table 5 jcm-13-01844-t005:** Postoperative complications in patients receiving the high-protein liquid diet prepared in the hospital and those receiving commercial polymeric formulas.

Complications	Hospital-Based High-Protein Liquid Diet (*n* = 87)	Commercial Formulas (*n* = 62)
No complications	73 (83.9)	60 (96.7)
Postoperative wound necrosis	4 (4.6)	–
Oral cutaneous fistula	4 (4.6)	–
Impaired healing of skin flaps	2 (2.3)	–
Gastrointestinal perforation following PEG tube insertion	–	1 (1.6)
Perioperative death	4 (4.6)	1 (1.6)

Data are presented as *n* (%) of patients. PEG—percutaneous endoscopic gastrostomy.

## Data Availability

The data that support the findings of this study are available from the corresponding author, upon reasonable request.

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
