# Peer review of "The Effect of Hospital-Based Liquid Diet and Commercial Formulas on Laboratory Parameters and Postoperative Complications in Patients with Head and Neck Cancer"

_jcm, 2024, doi:10.3390/jcm13071844_

Round 1
Reviewer 1 Report
Comments and Suggestions for Authors The manuscript aim is was to compare the effect of polymeric formulas available commercially and high-protein liquid diet prepared in the hospital on laboratory parameters and postoperative complications in patients undergoing surgery for HNC and the authors succeeded in follow the aim by supporting itt with a specific study on 149 patients who underwent surgery for HNC between 2008 and 2017 The study was valuable as the authors taken in their considerations the patient and tumor characteristics, postoperative complications, and laboratory parameters measured at baseline and after surgery. The results were represented i readable manner by using table only but to be more specific figures will provide the reader the data in more clearer manner. The conclusion is clear and improving the aim.
Author Response
Dear Reviewer,
Thank you for this review. We agree that adding figures to describe the results will improve our manuscript. We have included figures illustrating the percentage change in laboratory parameters after surgery in both groups and, following the suggestion of another reviewer, correlation charts.
Reviewer 2 Report
Comments and Suggestions for Authors
This study has retrospectively compared the kitchen-made formula with the commercial formula in patients with head and neck cancers. It is an interesting study due to limited related ones. My concern is that the authors did not compare the energy and nutrients of these formulas. The composition of the feedings is important, and they can affect some outcomes. Moreover, I recommend sub-analysis on patients with normal and abnormal baseline hepatic enzymes.
Author Response
Dear Reviewer,
Thank you for this review. In section 2.2, we have described the characteristics of the nutrition formulas. The energy and macronutrient supply were in line with the principles of proper nutrition for hospitalized patients in Poland. Unfortunately, we did not plan a sub-analysis on patients with normal and abnormal baseline hepatic enzymes in our statistical analysis plan. Nonetheless, there were no significant differences in baseline levels of hepatic enzymes between the two groups
Reviewer 3 Report
Comments and Suggestions for Authors
Dear Authors,
In the manuscript entitled “The effect of hospital-based liquid diet and commercial formulas on laboratory parameters and postoperative complications in patients with head and neck cancer,” the authors study outlines the effects of two diets on head and neck cancer patients have been investigated. My overall evaluation in this article is negative.
1. It is necessary to explain more about commercial and hospital-based formulations. At the same time, its ingredients and what animal or plant it is made from should be fully explained.
2. Why are two different types of food containers used for gavage of patients in the hospital? Do gavage liquids have preservatives?
3. It seems that there is not enough evidence to mention this claim" This may indicate that commercial formulas may cause metabolic burden on the liver.) due to the fact that the death rate in people fed with hospital-based high-protein liquid diet was higher than in people fed with commercial formulas.
4. According to the location of the tumor, if viruses such as HPV are the cause, it should be mentioned.
5. It is necessary to use the graph to compare the groups.
6. Why is there no healthy control group in the article?
7. It is necessary to explain in this regard if patients have metastases to vital organs.
Author Response
In the manuscript entitled “The effect of hospital-based liquid diet and commercial formulas on laboratory parameters and postoperative complications in patients with head and neck cancer,” the authors study outlines the effects of two diets on head and neck cancer patients have been investigated. My overall evaluation in this article is negative.
Authors: Thank you, for this review and all comments.
It is necessary to explain more about commercial and hospital-based formulations. At the same time, its ingredients and what animal or plant it is made from should be fully explained.
Authors: Thank you for this comment. Unfortunately, we do not have such information. All nutrition interventions were conducted in accordance with the nutrition guidelines for hospitalized patients.
Why are two different types of food containers used for gavage of patients in the hospital? Do gavage liquids have preservatives?
Authors: We honestly don't understand this question, could you please explain to us what you mean?
It seems that there is not enough evidence to mention this claim" This may indicate that commercial formulas may cause metabolic burden on the liver.) due to the fact that the death rate in people fed with hospital-based high-protein liquid diet was higher than in people fed with commercial formulas.
Authors: This finding is based on the observation that patients receiving the commercial diet had significantly higher increases in ALT and AST. It is true that we observed a higher number of deaths in patients on the hospital-based high-protein liquid diet; however, these were perioperative deaths. Regardless, we have decided to delete this statement.
According to the location of the tumor, if viruses such as HPV are the cause, it should be mentioned.
Authors: Thank you for pointing this, in our Centre HPV is tested only for the middle and posterior pharynx. Our cases involved the front of the mouth
It is necessary to use the graph to compare the groups.
Authors: Thank you for this suggestion, we have added graph with group comparison.
Why is there no healthy control group in the article?
Authors: We do not see the possibility of including a healthy control group in this study, as we are investigating the impact of enteral nutrition in patients undergoing surgery
It is necessary to explain in this regard if patients have metastases to vital organs.
Authors: The included patients had no metastases to vital organs, as this is a contraindication to surgery.
Reviewer 4 Report
Comments and Suggestions for Authors
1. On the method section line 154-154, it is stated that "Significant differences between the means in 2 groups were determined using the Student t-test". However, some of the analysis should be conducted using paired t-test or wilcoxon test for non-parametric data. Thus, please mention on the method section whether paired t-test or wilcoxon test were used.
2. For table 2, as the data is presented using median, please use mann-whitney test or wilcoxon test for statistical comparison if this had not been conducted. Please do not use t-test for non-parametric data.
3. Scatterplots figure for table 4 would be a good addition for this manuscript
I do not find any other issues with the manuscript
Author Response
Dear Reviewer, thank you for this review and valuable comments.
1. On the method section line 154-154, it is stated that "Significant differences between the means in 2 groups were determined using the Student t-test". However, some of the analysis should be conducted using paired t-test or wilcoxon test for non-parametric data. Thus, please mention on the method section whether paired t-test or wilcoxon test were used.
Authors: You are right, we also used paried test, we added it to the methodology.
2. For table 2, as the data is presented using median, please use mann-whitney test or wilcoxon test for statistical comparison if this had not been conducted. Please do not use t-test for non-parametric data.
Authors: You are right; providing medians was confusing. We have changed the medians to means with standard deviations, as we are using parametric tests since we observed normality in the distribution.
3. Scatterplots figure for table 4 would be a good addition for this manuscript
Authors: Thank you for this suggestion, we have added scatterplots for table 4.
I do not find any other issues with the manuscript
Authors: Thank you.
Reviewer 5 Report
Comments and Suggestions for Authors
This manuscript presents a comparison of different approaches to the feeding of HNC patients in hospital settings. The motivation is clear and the method and data are adequately described. However, before the large number of "NS" findings can be correctly interpreted, the reader needs to know their context; i.e., for these variables, what were the minimum necessary sample sizes required to produce a significant result? Were these "NR's" consequences of inadequate samples with large data spreads? Resolution of this issue will greatly enhance the usefulness of this report.
Author Response
Dear Reviewer,
Thank you for this valuable opinion. The reason why we did not perform a sample size analysis is due to the retrospective nature of the study, and the number of patients enrolled was determined based on their current availability and the inclusion criteria.
According to publication by Kim and Seo retrospective studies use statistical power rather than the calculation of sample sizes and we call these 'post hoc power analyses'. (https://www.ncbi.nlm.nih.gov/pmc/articles/PMC3758995/)
Yesterday, we performed a power analysis and obtained various levels of power for different parameters. For example, the power for hemoglobin levels before surgery was 4.9%, and we need 6279 patients to achieve a power of 80%. In our study, we included 149 patients over a 9-year period. According to our calculations, any difference may be statistically significant if we have a sufficiently large group.
Zhang et al. in their paper wrote: "Power analysis is an indispensable component of planning clinical research studies. However, when used to indicate power for outcomes already observed, it is not only conceptually flawed but also analytically misleading. Our simulation results show that such power analyses do not indicate true power for detecting statistical significance, since post hoc power estimates are generally variable in the range of practical interest and can be very different from the true power."
Therefore we not included power analysis in our study. If you have different opinion please let us know.
Round 2
Reviewer 4 Report
Comments and Suggestions for Authors
Thankyou for the revisions. I have no further comments. Please double check data numbers on galley proof to ensure data accuracy.
Author Response
Thank you so much for your time. We will check carefully all the proofs.
Reviewer 5 Report
Comments and Suggestions for Authors
Thank you for considering my request. However, I fear you may have missed the point. The purpose of a sample size calculation (under any conditions) and of the corresponding "power" is to reassure the reader that any findings of "NS" are reliably based on a theoretical possibility of observing a statistically significant difference, given the inherent variability of the data and do not relate to whether a finding of "p<.05" is "significant." Please address (in the manuscript) your rationale for asking your readers to accept your many findings of "NS."
Author Response
Dear Reviewer,
Thank you for your comment. We understand your point and have added an explanation to the limitations section accordingly.